# Amivantamab-Vmjw: A Novel Treatment for Patients with NSCLC Harboring EGFR Exon 20 Insertion Mutation after Progression on Platinum-Based Chemotherapy

**DOI:** 10.3390/biomedicines11030950

**Published:** 2023-03-20

**Authors:** Vishal Shah, Andrea McNatty, Lacey Simpson, Henry Ofori, Farah Raheem

**Affiliations:** 1Department of Pharmacy, Mayo Clinic, 5881 E Mayo Blvd, Phoenix, AZ 85054, USA; 2Seagen, Bothell, WA 85054, USA

**Keywords:** amivantamab-vmjw, lung cancer, EGFR, exon 20, NSCLC

## Abstract

Objective: This study is a comprehensive review of the clinical pharmacology, pharmacokinetics, efficacy, safety, and clinical applicability of amivantamab-vmjw for metastatic non-small cell lung cancer (NSCLC) with epidermal growth factor receptor (EGFR) exon 20 insertion (exon20ins) mutation. Data Synthesis: The literature search to identify clinical trials returned only the CHRYSALIS phase 1 study. In a phase I trial, amivantamab-vmjw was associated with an overall response rate (ORR) of 40% (95% CI, 29–51) in the EGFR exon20ins NSCLC patient population (n = 81) after platinum-based chemotherapy. There were 3 complete responses (CRs) and 29 partial responses (PRs). The median duration of response (DOR) was 11.1 months (95% CI, 6.9—not reached; NR). The median progression-free survival (PFS) was 8.3 months (95% CI, 6.5–10.9), and overall survival (OS) was 22.8 months (95% CI, 14.6—NR). Application to Clinical Practice: This review summarizes the pharmacology, clinical evidence, and use of amivantamab-vmjw for patients with locally advanced or metastatic NSCLC with EGFR exon20ins mutation. Conclusion: The FDA approval of amivantamab-vmjw, the first bispecific antibody to target the exon20ins mutation, represents an important advancement in the treatment of patients with NSCLC with limited effective treatment options. The initial findings of the CHRYSALIS trial demonstrate an overall tumor response benefit with an acceptable safety profile.

## 1. Introduction

Lung cancer is one of the most common types of cancer among both males and females, accounting for 25% of all cancer-related deaths [1,2,3]. NSCLC accounts for 85% of all diagnosed types of lung cancer [1,4]. NSCLC is biologically and molecularly diverse, and targeted therapies against oncogenic driver mutations have become pivotal in improving survival outcomes [1,5]. Due to the heterogeneous nature of NSCLC, over 600 EGFR mutation variants have been described [6]. Driver mutations within the EGFR exons 18–21 occur in up to 40% of NSCLC cases [6,7]. Point mutations in exons 18, 19, and 21 represent 85% of EGFR mutation-positive NSCLC cases and have been shown to be responsive to targeted treatment using EGFR-specific tyrosine kinase inhibitors (TKIs) [8,9]. The next most frequently occurring EGFR mutation is exon20ins, accounting for approximately 12% of cases.

EGFR exon20ins mutations are defined by in-frame insertions and duplications near the EGFR kinase domain. EGFR exon20ins mutations contain a modified active site that hinders the binding of currently approved EGFR TKI molecules, resulting in low response rates of <9%. [8,9] Due to this low response rate, the median OS in NSCLC with EGFR exon20ins mutations is only 16 months [10,11,12,13,14]. The absence of tumor response to standard EGFR TKIs and suboptimal efficacy of chemotherapy highlight the cruciality of identifying targeted therapy with activity against EGFR exon20ins mutations in NSCLC.

Amivantamab-vmjw presents a novel mechanism as a bispecific EGFR and mesenchymal epithelial transition (MET) receptor antibody with accelerated FDA approval in May 2021 for patients with metastatic NSCLC with EGFR exon20ins mutations who have progressed after platinum-based chemotherapy [15]. The accelerated approval was based on the positive results of overall response rate as well as the duration of response demonstrated by an open-label, phase I clinical trial [8]. In this article, we describe the pharmacology, clinical efficacy, and safety of amivantamab-vmjw for treatment of EGFR exon20ins mutation-positive NSCLC.

## 2. Pharmacology

Acquired resistance to standard of care treatments can develop in NSCLC due to mutations in EGFR and activation of the MET pathway. EGFR and MET pathways are required for tumor cell proliferation and growth. The mutations in the EGFR protein activate a chain of events that constitutes subsequent growth and survival signaling in cancer cells [7,9]. EGFR mutations develop in up to 85% of patients with NSCLC [8]. Driver mutations in the exons of the EGFR tyrosine kinase domain have been evolving targets for NSCLC therapies. First-, second-, and third-generation EGFR TKIs have been established as standard of care treatment for NSCLC with EGFR mutations except for exon20ins mutations. EGFR exon20ins mutation is a distinct subtype of EGFR oncogenic mutations that has been shown to be unresponsive to third-generation EGFR TKIs [8,9]. The use of EGFR TKIs can also induce MET receptor amplification, which bypasses the EGFR pathway, causing signaling downstream to promote cancer cell proliferation and resistance to standard of care treatments [16].

Amivantamab-vmjw demonstrates a novel mechanism of action as a fully humanized immunoglobulin G1-based bispecific antibody that targets the domains of EGFR and MET. In vivo and in vitro studies of amivantamab-vmjw demonstrated a dose-dependent induction of EGFR and MET downregulation, antiproliferative effect, and antibody-dependent cellular cytotoxicity [15]. Antitumor effects were shown with both Fc receptor-independent and -dependent mechanisms. Fc-independent functions block EGFR and MET signaling via ligand binding and receptor inactivation. Amivantamab-vmjw binding to the exon20ins-mutated EGFR tyrosine kinase receptor results in degradation of EGFR and MET receptors [7,9,17]. The Fc-dependent mechanism of action plays in important role in amivantamab-vmjw pharmacological activity through antibody-dependent cellular cytotoxicity by activation of natural killer cells, macrophages, and monocytes. A subsequent cytokine and chemokine release is triggered along with downmodulation of EGFR and MET receptors through trogocytosis with Fc interaction [7].

Amivantamab-vmjw is a bispecific monoclonal antibody produced from two different bivalent parental antibodies. The EGFR arm is a derivative of zalutumumab, which has a conformational epitope of EGFR domain III, but is different from epitopes of cetuximab and panitumumab. The epitope on the MET-binding arm blocks the HGF ligand but is dissimilar from onartuzumab [18]. Amivantamab-vmjw binds to extracellular domains of EGFR and MET receptors with a dissociation constant of 1.43 and 0.04 nM, respectively [19].

## 3. Pharmacokinetics and Pharmacodynamics

The phase 2 clinical trial recommended dose of amivantamab-vmjw was selected based on serum EGFR and MET target saturation in addition to the determined preclinical concentration of 168 ug/mL [8]. Saturation of EGFR and MET circulating targets was achieved with doses above 700 mg. Dosing was determined by weight utilizing population kinetics to minimize pharmacokinetic variability, leading to recommendations of 1400 mg with weight ≥ 80 kg and 1050 mg for <80 kg. The volume of distribution and clearance of amivantamab-vmjw increased due to body weight escalation, which is attributed to the two-tiered weight-based dosing. Exposure was similar when comparing the 1050 mg dose at <80 kg and 1400 mg for those who weighed ≥ 80 kg. Pharmacokinetic results of amivantamab-vmjw showed no clinically significant differences regarding age sex, race, creatinine clearance (CrCl) ≥ 29 mL/min, or mild hepatic impairment. Amivantamab-vmjw was not investigated in CrCl < 29 mL/min or in moderate (bilirubin 1.5 to 3 times upper limit normal (ULN)) or severe (total bilirubin > 3 times ULN) hepatic impairment. Mean linear clearance was determined to be 0.36 L/d and have a terminal half-life of 11.3 (+/− 4.53) days [8,15].

## 4. Clinical Trial Efficacy

The safety and efficacy of amivantamab-vmjw was evaluated in the CHRYSALIS trial, a phase I, open-label, multicenter, dose escalation and dose expansion study that included patients with advanced NSCLC with EGFR exon20ins mutation. Patients included were adults ≥ 18 years, had metastatic or unresectable NSCLC, ECOG status of ≤1, and had progressed on standard of care therapy. Patients with untreated or active brain metastasis were not included in the trial. Previously treated and asymptomatic brain metastases were allowed. Patients in the dose expansion phase had documented EGFR mutations or MET mutations and quantified disease defined by RECIST 1.1 criteria. Tissue or central next-generation sequencing (NGS) circulating tumor DNA (ctDNA) testing was utilized to detect EGFR exon20ins mutations.

The dose escalation phase was a 3 + 3 design that assessed amivantamab-vmjw dosing weekly for the first 28 days followed by biweekly dosing. Patients were divided into six different dose cohorts for dose escalation. The analysis presented here includes patients with NSCLC EGFR exon20ins mutations (cohort D). The main objective of the dose escalation phase was to determine the maximum tolerated dose for the phase II portion of the study. For phase II, dose expansion, six cohorts were identified based on EGFR and MET mutations or amplifications and previous therapy status. The primary objective for the phase II portion of the study was ORR. Secondary endpoints included DOR; clinical benefit rate (CBR), defined as PR or CR or stable disease for ≥11 weeks; PFS; and OS. Treatment with amivantamab-vmjw extended until disease progression, unacceptable toxicity, or withdrawal of consent.

From May 2016 to June 2020, 362 patients were enrolled in the study. Patients comprised in the safety population included those with EGFR exon20ins mutations that progressed on platinum-based therapy and were treated at the phase II dose (n = 114). A total of 258 patients were given a dosage of 1050 mg for patients weighing < 80 kg and 1400 mg for patients ≥ 80 kg given once weekly for the first four weeks, then biweekly starting at week five. The efficacy population consisted of the first 81 patients with EGFR exon20ins mutations that had received previous platinum-based chemotherapy and had ≥3 disease assessments at data cutoff. Data were assessed through 8 October 2020 for this population. The median age of the efficacy population was 62 years (42–84 years), and 59% were women, 49% were of Asian descent, 53% were nonsmokers, and 95% had adenocarcinoma. The median number of previous lines of therapy was two (range 1–7). All patients had previously received platinum-based chemotherapy, 25% received prior EGFR TKIs, and 46% had received prior immuno-oncology therapies. Approximately 22% of patients had treated brain lesions at enrollment.

At median follow up of 9.7 months, the ORR was 40% (95% CI, 29–51) with 3 confirmed CR and 29 PR outcomes, as evaluated by a blinded independent central review. The investigator-assessed ORR was 36% (95% CI, 25–47). The median DOR was 11.1 months (95% CI, 6.9-NR). CBR was observed in 74% of patients (95% CI, 63–83). Median PFS was 8.3 months (95% CI, 6.5–10.9). Median OS was 22.8 months (95% CI, 14.6-NR). The overall survival endpoint is currently immature. Efficacy endpoints are summarized in Table 1 [8].

## 5. Safety

The National Cancer Institute Common Terminology Criteria for Adverse Events (CTCAE), version 4.03 was applied for grading AEs. The most frequent AEs observed in the CHRYSALIS trial were rash (84%), infusion-related reactions (IRR (64%)), paronychia (50%), musculoskeletal pain (47%), dyspnea (37%), nausea (36%), fatigue (33%), edema (27%), stomatitis (26%), cough (25%), constipation (23%), and vomiting (22%). Rash including acneiform dermatitis, paronychia, stomatitis, and diarrhea are associated with on-target anti-EGFR TKIs activity, while peripheral edema is associated with on-target anti-MET activity. Grade ≥ 3 AEs were reported in 16% of patients including rash (4%), IRRs (3%), and neutropenia (3%). All grade interstitial lung disease and pneumonitis occurred in 4% of patients [8,15]. The types and rates of AEs are summarized in Table 2.

Dose reduction due to AEs occurred in 13% of patients of which rash was the most frequent cause (10%). Adverse events causing permanent discontinuation of treatment occurred in ≥11% of patients and included pneumonia, IRRs, pneumonitis/interstitial lung disease, pleural effusion, and rash [8,15].

Overall, 66% of patients treated with amivantamab-vmjw resulted in IRRs, causing infusion interruptions in 59% of patients. IRRs developed almost exclusively on cycle 1 day 1 (65%) or day 2 (3.4%). IRRs rarely developed during subsequent cycles (1.1%). Of the reported IRRs, 97% of them were grade 1–2, 2.2% were grade 3, and 0.4% were grade 4. There were no predisposing factors identified. To mitigate the development of IRRs, the first dose was split over two days (cycle 1 day 1 and 2). Prophylactic premedications were required (Table 3). The median time for onset was 1 h (range, 0.1 to 18 h) after the start of the infusion. Dose modifications due to IRRs occurred in 62% of patients, and 1.3% terminated treatment due to IRRs. Signs and symptoms of IRR are listed in Table 2.

If patients experienced infusion reactions during subsequent cycles, infusion was interrupted until symptom resolution and resumed at 50% of the previous infusion rate [8]. Management of IRRs included infusion interruptions, reducing the rate of administration, and administration of supportive care, including antihistamines, antipyretics, glucocorticoids, epinephrine, and meperidine, which is shown in Table 3 [8].

EGFR inhibitors have been shown to cause skin-related toxicities [17,20]. Prevention of the EGFR inhibitor induced rash-related AEs during the CHRYSALIS trial, which included using broad-spectrum sunscreen and alcohol-free emollients and limiting exposure to sunlight [8]. Management of EGFR inhibitor-induced rash consisted of antihistamines, corticosteroids, topical antibiotics (clindamycin), systemic antibiotics (doxycycline 100 mg BID or minocycline 100 mg BID), antiseptic soaks, and local corticosteroids [8].

## 6. Dosing and Administration

The suggested dosing for amivantamab-vmjw is based on the patient’s baseline body weight. Subsequent body weight changes do not require further dose adjustments. For patients weighing < 80 kg, the recommended dose is 1050 mg (3 vials) [15]. For patients weighing ≥ 80 kg, the recommended dose is 1400 mg (4 vials) [15]. For the first 4 weeks, amivantamab-vmjw is administered weekly. Starting at week 5, amivantamab-vmjw is administered every 2 weeks (Table 4). Administration of an antihistamine (diphenhydramine 25 to 50 mg) and antipyretic (acetaminophen 650 to 1000 mg) is required prior to all doses of amivantamab-vmjw. A glucocorticoid (dexamethasone 10 mg or methylprednisolone 40 mg) administration is mandatory prior to cycle 1, days 1 and 2 of amivantamab-vmjw infusion (Table 3) [15]. Dose modifications for toxicity are outlined in Table 5, and modification guidance for adverse reactions is found in Table 6.

Amivantamb-vmjw is administered as an intravenous infusion using an in-line low protein-binding polyethersulfone 0.2-micron filter. To minimize infusion reactions, the first dose is split over day 1 and day 2. Additionally, the administration tubing set is primed with 15–25 mL of diluent (dextrose or saline) to slow initial exposure to amivantamab-vmjw. Another strategy to minimize the risk for IRRs is withdrawing and discarding 10 mL of blood prior to flushing the catheter with dextrose or saline at the end of drug administration to avoid rapid infusion of residual amivantamab-vmjw. Additionally, do not flush when infusion must be interrupted for toxicity [21].

While there are no data evaluating amivantamab-vmjw administration vis a peripheral versus central line, it is suggested to infuse amivantamab-vmjw via a peripheral line in weeks 1 and 2 when IRRs are most frequent. Administration via a peripheral line allows for slower infusion, thus reducing the risk for IRRs. A central line can be utilized following the first two weeks of therapy. It is also recommended to consider dose preparation as close to administration time as possible to allow for potential extended infusion times if IRRs occur [22]. Recommended administration rates vary based on dose and treatment cycles and are summarized in Table 7.

## 7. Cost

Amivantamab-vmjw is available as a single-dose 350 mg/7 mL (50 mg/mL) vial. The average wholesale price for each vial is around USD 3296 [23]. For patients weighing less than 80 kg, the dose is three vials, and the cost is approximately USD 9888. For patients weighing more than or equal to 80 kg, the dose is four vials, the cost is approximately USD 13,184. The CHRYSALIS study patients were treated for the median duration of 5.6 months, which amounts to approximately USD 138,432 to USD 184,576 for six treatment cycles. Center for Medicare and Medicaid Services (CMS) has established a new HCPCS level II code J9061 in 2 mg increments for billing purposes.

## 8. Application to Clinical Practice

Patients with EGFR exon20ins mutations in NSCLC represent 10–12% of all EGFR mutations. Prognosis is usually poor, and effective treatment options have been very limited. Efficacy with EGFR TKIs, such as erlotinib or osimertinib, are limited by low-binding affinity at the tyrosine kinase domain of Exon 20. Upfront platinum-based chemotherapy is offset by a median OS of 12.5 months in the relapsed or refractory setting, making amivantamab-vmjw an attractive choice for second- or third-line therapy [24].

Recently, an oral EGFR exon20ins mutation-directed TKI, mobocertinib, was approved by the FDA for treatment of advanced NSCLC. The ORR in 114 patients with NSCLC harboring EGFR exon20ins mutations who had made progress on platinum-based chemotherapy was 28% (95% CI, 20–37) [25]. Approximately 46% of patients exhibited treatment-related AEs of grade ≥ 3. Diarrhea and rash were the most frequently reported AEs, at 90% (21% grade ≥ 3) and 45%, respectively [25].

Amivantamab-vmjw presents a manageable safety and toxicity profile with IRRs being the most common adverse effect. IRRs can be mitigated with splitting of the dose over two days during week 1 and administration of antihistamines, antipyretics, and glucocorticoids prior to infusion. IRRs were managed by infusion interruption and reduction of the infusion rate. The incidence of IRRs decreases dramatically from day 1 to day 2 and in subsequent weeks with reported rates of 65% on week 1 day 1, 3.4% on day 2, 0.4% on week 2, and 1.1% for all ensuing infusions. Administration of pre-infusion glucocorticoids is required on week 1 days 1 and 2 but optional in subsequent cycles. Continuation of pre-infusion antihistamines and antipyretics is recommended for all cycles. Practical recommendations include administering amivantamab-vmjw through a peripheral line on days 1 and 2, scheduling infusion appointments early in the morning, preparing amivantamab-vmjw close to the scheduled administration appointment to allow for extended infusion time during appropriate working hours of the infusion center, and having cardiopulmonary resuscitation equipment and medication needed to treat infusion reactions readily available.

Dermatologic reactions were commonly seen in CHRYSALIS trial with 86% of patients experiencing rash of any grade, including dermatitis acneiform, which is one of the most common on-target dermatological AEs associated with EGFR inhibitors. The median time for the onset of rash was 14 days (range, 1 to 276 days). Topical steroids and/or topical antibiotics and oral antibiotics are typically recommended for EGFR inhibitors’ mediated rash. Other interventions include applications of topical emollients and broad-spectrum UVA/UVB sun lotion during treatment and for 2 months following discontinuation of treatment.

Amivantamab-vmjw was FDA approved based on early efficacy data that led to a Breakthrough Therapy designation. This was based on the investigator-assessed ORR of 41%, median DOR of 7 months, and CBR of 72%. This efficacy analysis is based on a small sample size of 81 patients [8]. Furthermore, patients with active or untreated brain metastases were excluded, and the benefit of amivantamab-vmjw is unknown in this patient population, which is a key limitation of this therapy. Efficacy analysis is ongoing for patients with EGFR exon20ins mutations in other disease states besides NSCLC, in patients who have not been previously treated with platinum-based chemotherapy, and in combination with other therapies. The National Comprehensive Cancer Network (NCCN) guidelines suggest (category 2A) either amivantamab-vmjw or mobocertinib for subsequent therapy for patients with exon20ins mutations who have progressed during or after initial systemic treatment including chemotherapy with or without immunotherapy [26]. Comparison of mobocertinib and amivantamab-vmjw is outlined in Table 8. After progression on amivantamab-vmjw or mobocertinib, the recommendation is to switch to the exon20ins mutation-targeted treatment that was not previously given. Ongoing clinical trials targeting exon20ins mutations are included in Table 9.

## 9. Conclusions

The FDA approval of amivantamab-vmjw, the first bispecific antibody to target EGRF exon20ins, represents an important development in the treatment of patients with advanced or metastatic NSCLC who have previously had limited effective treatment options. The early findings of the CHRYSALIS trial not only show a benefit in overall response rate, but also a relatively tolerable safety and toxicity profile. In addition, the efficacy results show durable responses in patients who do benefit from this therapy. CHRYSALIS is an ongoing study that is evaluating the effectiveness and safety of amivantamab-vmjw as monotherapy and in combination with other therapies in NSCLC and other solid malignancies harboring EGFR exon20ins mutations. Long-term follow up is necessary to confirm efficacy and safety in the study population and in real-world patients.

## Figures and Tables

**Table 1 biomedicines-11-00950-t001:** Efficacy Outcomes of Amivantamab-vmjw for NSCLC Harboring EGFR Exon20ins Mutations [8].

Response	CR No. (%)	PR No. (%)	SD No. (%)	PD No. (%)	ORR % (95% CI)	NE No. (%)	mPFS(95% CI)	mOS(95% CI)
Efficacy population (n = 81)	3 (4)	29 (36)	39 (48)	8 (10)	40 (29–51)	2 (2)	8.3 mo (6.5–10.9)	22.8 mo (14.6-NR)

CR, complete response; PR, partial response; SD, stable disease; PD, progressive disease; ORR, overall response rate; NE, not evaluable; mPFS, median progression-free survival; mo. Months; mOS, median overall survival; NR, not reached; NSCLC, non-small cell lung cancer.

**Table 2 biomedicines-11-00950-t002:** Reported Adverse Events with Amivantamab-vmjw in CHRYSALIS [8].

Common AE	Safety Population (n = 114) No. (%)				Patients Treated with the Phase II Dose (n = 258) No. (%)			
	Total	Grade 1	Grade 2	Grade ≥ 3	Total	Grade 1	Grade 2	Grade ≥ 3
Rash	98 (86)	43 (38)	51 (45)	4 (4)	202 (78)	101 (39)	94 (36)	7 (3)
Infusion-related reactions	75 (66)	9 (8)	63 (55)	3 (3)	167 (65)	21 (8)	140 (54)	6 (2)
Paronychia	51 (45)	28 (25)	22 (19)	1 (1)	104 (40)	50 (19)	51 (20)	3 (1)
Constipation	27 (24)	18 (16)	9 (8)	0	58 (23)	36 (14)	22 (9)	0
Dyspnea	22 (19)	12 (11)	8 (7)	2 (2)	52 (20)	28 (11)	13 (5)	11 (4)
Nausea	22 (19)	17 (15)	5 (4)	0	(55) 21	40 (16)	14 (5)	1 (0.4)
Vomiting	12 (11)	10 (9)	2 (2)	0	29 (11)	22 (9)	6 (2)	1 (0.4)
Fatigue	21 (18)	15 (13)	4 (4)	2 (2)	47 (18)	29 (11)	16 (6)	2 (1)
Stomatitis	24 (21)	11 (10)	13 (11)	0	50 (19)	33 (13)	17 (7)	0
Cough	16 (14)	11 (10)	5 (4)	0	40 (16)	25 (10)	15 (6)	0
Myalgia	14 (12)	12 (11)	2 (2)	0	28 (11)	23 (9)	5 (2)	0

**Table 3 biomedicines-11-00950-t003:** Prophylactic Premedications for Infusion-Related Reactions and Antiemetics [8].

Required Premedications			
Medication	Dose and Route of Administration	Dosing Window Prior to Drug Administration	Cycle/Day
Glucocorticoid ^a^	Dexamethasone 10 mg IV or Methylprednisolone 40 mg IV	45 to 60 min	Cycle 1 day 1 and cycle 1 day 2 only
Antihistamine ^b^	Diphenhydramine 25–50 mg	IV, 15 to 30 min PO, 30 to 60 min	All cycles
Antipyretic ^b^	Acetaminophen 650–1000 mg	IV, 15 to 30 min PO, 30 to 60 min	All cycles
**Optional Premedications**			
Glucocorticoid ^a^	Dexamethasone 10 mg or Methylprednisolone 40 mg	IV, 45 to 60 min PO, 60 to 90 min	Cycle 1 day 8 and beyond
H2 receptor antagonist	Famotidine 20 mg IV	15 to 30 min	Any cycle
Antiemetics	Ondansetron 8 mg PO or 16 mg IV	15 to 30 min	Any cycle

PO, oral; IV, intravenous. ^a^ beginning with cycle 1 day 8, optional pre-dose steroids were administered if clinically indicated for patients who experienced an infusion-related reaction on cycle 1 day 1 or cycle 1 day 2. ^b^ Required at all doses.

**Table 4 biomedicines-11-00950-t004:** Amivantamab-vmjw Dose Based on Body Weight and Dosing Schedule [15].

Baseline Body Weight (kg)	Dose (mg)	Number of 350 mg/7 mL Vials
<80 kg	1050 mg	3
≥80 kg	1400 mg	4
Dosing Schedule		
Cycle 1–4	Week 1 ^a^	Split infusion on C1D1 and C1D2
	Weeks 2–4	Infusion on Day 1
Cycles 5+		Every 2 weeks starting at week 5

^a^ C1D1: Cycle 1 Day 1; C1D2: Cycle 1 Day 2.

**Table 5 biomedicines-11-00950-t005:** Amivantamab-vmjw Dose Reduction Levels for Adverse Reactions [15].

Body Weight at Baseline	Dose Level	Dose
Less than 80 kg	0	1050 mg
	−1	700 mg
	−2	350 mg
Greater than or equal to 80 kg	−3 0 −1 −2 −3	Discontinue 1400 mg 1050 mg 700 mg Discontinue

**Table 6 biomedicines-11-00950-t006:** Amivantamab-vmjw Dose Modifications for Adverse Reactions [15].

Toxicity	Severity (CTCAE Grade *)	Dose Modification
Dermatologic	2	Supportive care treatment Reassess after 2 weeks Dose reduction if rash does not improve
	3	Hold amivantamab Supportive care management Resume with dose reduction ≤ Grade 2 Discontinue if no improvement within 2 weeks
	4	Permanently discontinue
	Severe blistering, bullous, exfoliation of skin (including TENS)	Permanently discontinue
Infusion related reactions	1 or 2	Stop infusion and monitor for symptom resolution Supportive care management Resume at 50% initial infusion rate Escalate infusion rate after 30 min if no additional symptoms occur Corticosteroid premedication for subsequent infusions ^a^
	3	Stop infusion and monitor for symptom resolution Supportive care management Resume at 50% initial infusion rate Escalate infusion rate after 30 min if no additional symptoms occur Corticosteroid premedication for subsequent infusions ^a^ Permanently discontinue with recurrent Grade 3
	4	Permanently discontinue
Interstitial lung disease/Pneumonitis	Any	Hold dose if suspected Permanently discontinue if confirmed
Other	3	Hold dose until recover ≤ Grade 1 or baseline Resume same dose if recovery within 1 week Resume with dose reduction if recovery within 1–4 weeks Permanently discontinue if no recovery in 4 weeks
	4	Hold dose until recover ≤ Grade 1 or baseline Resume with dose reduction if recovery within 4 weeks Permanently discontinue if no recovery in 4 weeks Permanently discontinue with recurrent Grade 4

TENS: toxic epidermal necrolysis. * NCI Common Terminology Criteria for Adverse Events (CTCAE) severity Grade 1–5 scale. ^a^ Dexamethasone 10 mg IV or methylprednisolone 40 mg IV 45–60 min prior to amivantamab administration.

**Table 7 biomedicines-11-00950-t007:** Infusion Rates of Amivantamab-vmjw Administration [15].

1050 mg Dose (for <80 Kg)			
Week	Dose (Prepared in 250 mL Bag)	Initial Infusion Rate	Subsequent Infusion Rate
1 (split dose)			
Day 1	350 mg	50 mL/h	75 mL/h
Day 2	700 mg	50 mL/hr	75 mL/h
2	1050 mg	85 mL/h	
3+	1050 mg	125 mL/h	
**1400 mg Dose (for ≥80 Kg)**			
Week	Dose (prepared in 250 mL bag)	Initial Infusion Rate	Subsequent Infusion Rate
1 (split dose)			
Day 1	350 mg	50 mL/h	75 mL/h
Day 2	1050 mg	35 mL/h	50 mL/h
2	1400 mg	65 mL/h	
3+	1400 mg	125 mL/h	

**Table 8 biomedicines-11-00950-t008:** FDA-Approved Treatments for Locally Advanced or Metastatic NSCLC with EGFR Exon20ins Mutations in the Recurrent Setting [8].

	CHRYSALIS (Amivantamab-Vmjw) (n = 81)	Study 101 (Mobocertinib) (n = 114)
ORR	40% (95% CI, 29–51%): 3.7% CR; 36% PR	28% (95% CI, 20–37) all PR
DOR	11.1 mo (95% CI 6.9, NE)	17.5 mo (95% CI, 7.4–20.3; n = 32/114)
DOR ≥ 6 mos	63%	59%
mOS	22.8 mo	24 mo
mPFS	8.3 mo	7.3 mo

ORR, overall response rate; CR, complete response; DOR, duration of response; mOS, median overall survival; mPFS, median progression-free survival; NSCLC, non-small cell lung cancer; PR, partial response.

**Table 9 biomedicines-11-00950-t009:** Ongoing Clinical Trials of Exon20ins Mutation-Targeted Therapy in NSCLC [27,28,29,30,31,32,33].

Trial ^a^	Investigational Intervention(s)	Phase	Allocation/Design	NSCLC Condition
PAPILLON (NCT04538664)	Amivantamab-vmjw and carboplatin-pemetrexed compared with carboplatin-pemetrexed	3	Randomized, open-label	EGFR exon20ins mutation
MARIPOSA-2 (NCT04988295)	Amivantamab-vmjw and lazertinib in combination with platinum-based chemotherapy compared with platinum-based chemotherapy alone	3	Randomized, open-label	EGFR-mutated Osimertinib failure
EXCLAIM (NCT04129502)	TAK-788 (mobocertinib) compared with platinum-based chemotherapy	3	Randomized, open-label	EGFR Exon 20 insertion mutation
CLN-081 (NCT04036682)	CLN-081	1/2a	N/A, open-label	EGFR Exon 20 insertion mutation
Poziotinib (NCT03066206)	Poziotinib	2	N/A, open-label	Mutant advanced solid tumors EGFR or HER2
ZENITH20 (NCT03318939)	Poziotinib	2	N/A, open-label	EGFR or HER2 Exon 20 insertion mutation
FAVOUR (NCT04858958)	Furmonertinib mesilate	1b	Randomized, open-label	EGFR Exon 20 insertion mutation

^a^ Clinicaltrials.gov identifier.

## Data Availability

Not applicable.

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
