# Peer review of "Amivantamab-Vmjw: A Novel Treatment for Patients with NSCLC Harboring EGFR Exon 20 Insertion Mutation after Progression on Platinum-Based Chemotherapy"

_biomedicines, 2023, doi:10.3390/biomedicines11030950_

Round 1

Reviewer 1 Report

1. Tables 6 and 5 are mixed up.

2. There are no references in the text of the article to tables 7-9.

3. It is interesting to compare the data from different clinical studies shown in Table 9 on the effectiveness of Amivantamab-vmjw and other drugs.

Author Response

1) Tables 5 and 6 are fixed. 

2) References to tables 7-9 added/referenced in the body.

3) We wanted to be inclusive of all other trials related to Exon 20 in our review.  

Reviewer 2 Report

This is a well-organized review with detailed information about the clinical study of Amivantamab-vmjw. There are no special issues to be raised. Two minor issues are 1. It would be recommended the physicochemical properties of the drug product be included in a separate paragraph. 2. It would be better if the methods how to confirm the biomarkers are described in the paragraph on inclusion criteria.

Author Response

Thank you for your review and comments.

1) We have added a short paragraph describing the physiochemical properties.

2) This is described on page 3, first paragraph. To make it clear, we have added that central next generation sequencing (NGS) was utilized.

Reviewer 3 Report

This manuscript, written by Dr. Shah et al., review type, with the title of "Amivantamab-vmjw: A Novel Treatment for Patients with NSCLC Harboring EGFR Exon 20 Insertion Mutation After Progression on Platinum-Based Chemotherapy" summarizes the most relevant findings of the CHRYSALIS phase 1 study.

The presence of a characteristic mutation in the epidermal growth factor receptor (EGFR) defines a subset of patients with non-small cell lung cancer (NSCLC) who are likely to have a favorable response to EGFR tyrosine kinase inhibitors (TKIs).

This manuscript focuses on uncommon EGFR mutations, the EGFR exon 20 insertion mutations. For patients with EGFR exon 20 insertion-mutated NSCLC that have progressed on chemotherapy (either with or without immunotherapy), the suggested treatment is either amivantamab or mobocertinib as a later-line option.

Amivantamab is a bispecific EGFR and MET receptor antibody that is approved by the US Food and Drug Administration (FDA) for patients with locally advanced or metastatic NSCLC with EGFR exon 20 insertion mutations whose disease has progressed on or after platinum-based chemotherapy

The manuscript is well written, and it easy to read, and understand.

Comments:

1) Could you please make a figure or several figures showing the pathological mechanisms of EGFR mutation, with pathway activation, and drug targets. This may help the reader who is less familiar with this particular type of mutation in lung cancer.

You may refer to the following article:

Vyse, S., Huang, P.H. Targeting EGFR exon 20 insertion mutations in non-small cell lung cancer. Sig Transduct Target Ther 4, 5 (2019). https://doi.org/10.1038/s41392-019-0038-9

2) In the abstract, from line 14, the results are shown. This data comes from Table 1 and associated paragraph. Could you please add the number of patients in the abstract?

3) In Table 1, the efficacy populations is 81. But 3+29+39+8 is 79 patients.

4) Could you please make a table showing the clinicopathological characteristics of the 81 patients? Is the series heterogeneous? Do the cases with better response had any factor known to be associated with favorable prognosis?

5) Could you please describe the definition of the ORR?

6) Do the study had a control group? Do patients without this treatment, but with EGFR exon 20 insertion mutation, have worse prognosis than the patients treated with Amivantamab-vmjw?

7) Could you please expand the description of the relationship between EGFR and MET? How related are both pathways?

8) The authors made a literature search, but only one study was found. So, this manuscript is a review of that study only?

Author Response

Thank you for your review and comments.

1) It is difficult to create similar figures within 5 days of turn around time. I have attempted to create 1 figure which is attached. We do not have tools or funding to retain professional services.

2) N=81 added to abstract

3) 2 patients were not evaluable per study. I have added this informatio to the table.

4) Clinicopathological characteristics were not provided in the CHRYSALIS trial. The CHRYSALIS phase I study can be considered heterogenous based on the baseline patient demographics including patients with different metastases, previous systemic therapies, NSCLC subtypes, etc, but there is not concrete information provided to determine if the series is heterogenous. Information regarding if cases with better responses were associated with favorable prognosis was not specifically noted or provided in the study.

5) ORR added to table 8

6) CHRYSALIS was a phase I clinical trial that had no control arm. In CHRYSALIS patients with NSCLC and EGFR exon 20 insertion mutation were not randomized to amivantamab vs standard of care chemotherapy and a confirmatory, randomized phase III trial is awaited as current FDA accelerated approval of amivantamab is contingent upon reuslts of such confirmatory studies.

7) I added the relationship between EGFR and MET in the manuscript. The use of EGFR TKIs can also induce MET receptor amplification which bypasses the EGFR pathway causing signaling downstream to promote cancer cell proliferation and resistance to standard of care treatments.

8) Approval of this novel agent was based on only CHRYSALIS study. We have added a table describing other Phase 3 clinical trials that are ongoing. This review includes other aspects of this novel agent including to application to clinical practice based on our experience since the approval.
